# Experiences of People with Cardiovascular Disease during COVID-19 in Sweden: A Qualitative Study

**DOI:** 10.3390/ijerph20085588

**Published:** 2023-04-20

**Authors:** Catharina Sjödahl Hammarlund, Anna Norlander, Christina Brogårdh

**Affiliations:** 1Department of Health Sciences, Lund University, SE-22100 Lund, Sweden; anna.norlander@med.lu.se (A.N.); christina.brogardh@med.lu.se (C.B.); 2The PRO-CARE Group, Faculty of Health Sciences, Kristianstad University, SE-29188 Kristianstad, Sweden; 3Department of Neurology, Rehabilitation Medicine, Memory Disorders and Geriatrics, Skåne University Hospital, SE-22185 Lund, Sweden

**Keywords:** activities of daily living, cardiovascular conditions, COVID-19, mental health, physical health

## Abstract

Although people with cardiovascular conditions were subjected to more rigorous restrictions during the COVID-19 pandemic, there is limited knowledge of how the restrictions affected their lives and well-being. Thus, the aim of this study was to describe how people with cardiovascular conditions experienced their life situation and physical and mental health during the second wave of the pandemic in Sweden. Fifteen participants (median age 69 years; nine women) were individually interviewed, and data were analyzed with systematic text condensation. The findings revealed that some of the participants were fearful of contracting COVID-19 as their medical condition made them vulnerable. Additionally, the restrictions changed their daily routines and their ability to take part in social activities, as well as their access to specialized outpatient care (medical check-ups and physiotherapy). Although emotional and psychological distress were present, several participants found strategies that reduced their worries, such as exercising and meeting friends outdoors. However, some had adopted a more sedentary lifestyle and unhealthy diets. These findings indicate that healthcare professionals should provide individualized support to persons with cardiovascular diseases in order to find well-functioning emotion- and problem-focused strategies aimed at improving physical and mental health during crises such as pandemics.

## 1. Introduction

The World Health Organization already declared COVID-19 to be an international public health concern in March 2020 [1,2]. The disease was treacherous in that the symptoms of COVID-19 could be asymptomatic or mild for some, whereas others could suffer from serious life-threatening complications. In addition, it was found that COVID-19 was particularly dangerous for older people and persons with underlying chronic diseases [3,4]. Therefore, governments worldwide took several actions, such as lockdowns and putting people in quarantine in order to mitigate the spread of COVID-19 and reduce the number of cases and deaths [1,5,6]. The restrictions and home confinements significantly changed everyday life for many people [7,8,9]. The limited access to social activities led to various degrees of physical, psychological, and psychosocial distress [7,10,11]. Feelings of uncertainty and anxiety, as well as a fear of contracting the disease and dying, were also reported [12]. In Sweden, the legislation only allowed less rigorous restrictions to be applied during the pandemic [6,13]. The Public Health Agency of Sweden recommended physical distancing, frequent and careful handwashing, working from or studying at home when possible, avoiding interaction with people in various social contexts, and staying at home when having the slightest symptoms of infection [6]. People 70 years and older were given even stricter recommendations [14], as a majority (94%) of those who died from COVID-19 during the first wave were older [14,15]. However, the more lenient restrictions in Sweden allowed important infrastructures to be functioning, and transportation, schools for children, shops/malls, gyms, etc., were partly running. This reduced the negative effects on physical and mental health among the Swedish population as well as the socioeconomic consequences of the pandemic [6,13,16]. In retrospect, it was shown that a majority of people in Sweden were satisfied with life as a whole and with other important life domains, with the exception of contact with friends and sexual life during the first wave of the pandemic [17]. However, around 30% reported that life as a whole had deteriorated because of the pandemic. Having no children living at home, being middle-aged, having other sources of income than being employed, and having a chronic disease were associated with lower satisfaction with life as a whole [17]. A change in physical activity was also reported [16], and the highest odds of decreased physical activity were found in people 70 years of age or older. People who had decreased levels of physical activity also reported lower life satisfaction and aerobic capacity than the younger age groups [18].

Even though it is well known that restrictive changes in how one can live one’s life can have a negative impact on physical and mental well-being [5,7,19,20], there is limited knowledge regarding how vulnerable people have perceived and managed their situation following COVID-19. One group that deserves attention is people with cardiovascular diseases, which was the group with the highest death rate in 2021. People with cardiovascular conditions are often older and belong to a risk group for contracting COVID-19 [21,22]. Thus, the aim of this study was to describe how people with various cardiovascular conditions experienced their life situation and physical and mental health during the second wave of the COVID-19 pandemic in Sweden.

## 2. Materials and Methods

### 2.1. Study Design

This study has a qualitative design and was part of a larger project on how people experienced the restrictions following COVID-19 in Sweden [17,18].

### 2.2. Recruitment of Participants

Participants with various cardiovascular conditions were recruited from a larger cohort [17,18] who had responded to an online survey during the autumn of 2020. The online survey was hosted by Lund University and created in Research Electronic Data Capture (REDCap) program. The announcement was posted on a Facebook page and directed to three different regions of Sweden; two regions with large outbreaks of the pandemic (Stockholm and Gothenburg) and the southernmost part of Sweden (Scania). However, the invitation was also shared by Facebook users with people outside these areas. People aged 18 years or older, able to read and understand Swedish, were invited to participate in the survey, which included questions about socio-demographics, perceived life satisfaction, physical activity, and various diseases. A total of 1080 persons responded to the survey. Their residential communities were evenly spread between villages, towns, and cities. At the end of the survey, participants were asked if they wanted to participate in an interview concerning how they had experienced their life situation and physical and mental well-being during the COVID-19 pandemic in Sweden.

In total, 24 participants with cardiovascular conditions were interested in being interviewed. Of these, 9 were not available when contacted by telephone or e-mail. Thus, 15 persons (whereof nine were women) agreed to participate. They represented the three regions mentioned above, as well as other parts of Sweden. Their median (min–max) age was 69 (35–78) years. Eight of the participants were married/cohabiting. Two were diagnosed with heart failure, whereof one had a pacemaker, one had a congenital heart defect, one had heart fibrillation, and ten participants had high blood pressure. Other comorbidities reported were atherosclerosis, lung emboli, and asthma.

### 2.3. Ethics

All participants gave their informed consent to participate after receiving written and oral information. The study was approved by the Swedish Ethical Review Authority (Dno. 2020-02776), and the principles of the Declaration of Helsinki were followed.

### 2.4. Procedure

Data were collected digitally by two of the authors (C.B. and A.N.) using a semi-structured interview guide. The interviews were recorded (lasting on average 60 min, ranging between 45 to 75 min) and transcribed verbatim. Each interview started with the following: “I will now ask some questions about your life-situation, and your physical and mental health during the COVID-19-pandemic in Sweden”. The participants were helped to elaborate on each question using encouraging phrases such as “Can you give an example?” or “Can you elaborate on this subject, please?”.

### 2.5. Data Analysis

Data were analyzed using systematic text condensation as described by Malterud [23], a method that is inspired by Giorgi’s phenomenological theories. First, each transcript was read to obtain a general understanding of the data and to identify primary themes. Next, meaning units, i.e., parts of the text that contained information that was relevant to the research questions, were formulated into codes. During this phase, all three authors (C.B., A.N., and C.S.H.) worked independently to find various perspectives and nuances of the material. Thereafter, the coded data from the authors were integrated into one set of data, and duplicates were removed. The coded data were structured into categories by their conceptual representation, and the contents of the meaning units of each category were reviewed. Finally, descriptions of the essence of each category were formulated, and the categories were sorted into themes.

## 3. Results

The analysis generated one overall theme, “To continue living and managing one’s health during changed conditions”, and three comprehensive categories: (1) awareness of an exposed situation; (2) changes in daily life habits; and (3) curbing emotional reactions, with three elaborated subcategories, respectively (see Figure 1).

### 3.1. Awareness of An Exposed Situation

#### 3.1.1. Experiences of Being Vulnerable

Some of the participants did not fully agree that they belonged to a group of people considered to be at a higher risk of falling seriously ill due to their underlying condition. However, being perceived as vulnerable was causing worry among some, as it was not possible to know whether the disease would hit hard or if the symptoms would be mild. In addition, they felt negatively affected by the daily press conferences announcing how many people that had died.

… *If you know you have high blood pressure, um, both my parents suffered a stroke before they passed away, and high blood pressure is definitely a marker. Eh… So that, just knowing that blood pressure is high makes you feel stressed, and then the blood pressure rises even more*.P 2

Some participants with chronic heart disease experienced that their access to specialist outpatient care was significantly reduced or shut down as an effect of the pandemic, which gave rise to some concern regarding their own health.

…*my doctor does medical check-ups every year, but it* [the appointment] *was cancelled this year or rather last year, because then* [due to the pandemic] *they were forced to reallocate resources*.P 5

#### 3.1.2. Conflicting Feelings towards the Restrictions

Most participants expressed an overall intention to follow the restrictions to reduce the spread of the coronavirus, but also felt somewhat discriminated against by belonging to a “risk group” and having more severe restrictions. Seeing others who did not comply with the recommendations on social distancing could be provocative, especially if having made their own sacrifices. It was also perceived as stressful to constantly evaluate the risk or appropriateness of taking part in various activities. Some applied a more pragmatic approach and relied on their own judgment regarding necessary precautions.

*It always feels like you’re doing something wrong, and that’s what will be so nice when it’s over, that you never, that is, you won’t have to feel guilty all the time, for doing something, something that maybe you shouldn’t*.P 13

#### 3.1.3. Experiences of Falling Ill from COVID-19

Some of the participants believed that they had suffered from COVID-19, although not all had been tested. They felt ashamed to have caught the virus, but also safer now that they had already had the disease. As it was still at the beginning of the pandemic, information regarding complications was not yet fully known. People were expecting that the symptoms would be similar to the ordinary flu, but early information indicated that the lungs could be seriously affected, which was frightening and scary when suffering from cardiovascular diseases.

…*when I was lying in bed and I took the test on Tuesday, I had a positive result on Wednesday at 3.30 in the afternoon… eh … and I got a bit frightened as I have a condition… lately they say that if you are following your medication you don´t need to be scared, ….but you didn´t know that in the beginning* [of the pandemic]….P 12

Compared to previous experiences of recuperating from illness or surgery, the participants described that recovering from COVID-19 took much longer. Those who had caught the virus felt that the recovery time was tough and arduous.

*I was sick for about two weeks. I think I stayed at home for one and a half weeks and then went back to work, and then I was tired and fell ill again… or it could be that I never really had recovered, but I never understood that… It took more than a month and then I felt that I was in a really, very bad shape. … I have a very small slope leading up to the house and it was… I had to take breaks to manage walking up to the house*….P 13

### 3.2. Changes in Daily Life Habits

#### 3.2.1. Changes at Work and in Daily Routines

The participants described that changes according to the restrictions were quickly implemented at their workplaces. For example, the staff was split up into smaller groups at lunch or during coffee breaks; co-workers were considerate and careful to keep their distance, they washed their hands regularly, and they used hand sanitation. The digital meetings increased in number, which was perceived as stressful. It was not the same to meet, teach, or present via video conference. At the same time, it was easier to arrange meetings with people from all over the country.

*I feel that lately, there has been some kind of inflation in scheduling digital meetings, now people have realized that these meetings are very easy to arrange, and I think that my co-workers have become less disciplined*.P 13

There were also other positive aspects following the restrictions. Those who could work from home felt that it was an advantage not to commute. The home-based work also allowed for positive lifestyle changes, such as starting a healthy diet or changing one’s living arrangements.

…*what’s been positive* [during the pandemic] *has kind of been that there’s…* [pause] *the society has sort of gone down to a calmer pace in a way. And that pace kind of fits better with my pace, or my energy, or what to say*.P 5

The participants frequently mentioned how they had found solutions and routines for everything that needed to function despite the restrictions. Everyday activities such as shopping for food or clothes were more often done online or at different times of the day to avoid crowding. Taking control of their lives and keeping some forms of routines were perceived as important for well-being. 

*Just being able to keep the routines, that keeps the days containing something, I think it does a lot for your well-being*…P 1

#### 3.2.2. Changes in Social Interactions

The participants longed to have spontaneous meetings with colleagues and have coffee together. Being able to go to the workplace was important as you could see and hear other people. For some who were living alone, loneliness was particularly difficult to tackle. Input from others was important to balance their own thoughts.

…*I´ve always been comfortable with… being alone…But now, during the pandemic when, when you realise that social life has changed a lot then I have experienced the feeling of loneliness. I´ve felt lonely*.P 11

To enable social interaction despite the restrictions, the participants tried to meet with family and friends outdoors. As this required a great deal of planning, it often resulted in less frequent meetings. Being geographically close made it easier to meet, whereas those who had family and friends abroad were limited by travel restrictions. Although several participants expressed that they tried to keep in touch with family and friends digitally, they missed physical contact with others. Interacting with family, co-workers, or students via video calls did not provide the same benefits as meeting in real life.

*Because I’m the kind of person who likes to hug other people //… to just be able to put a hand on their knee and say “I understand how you feel”, trying to convey my feelings through my hand, I think that’s very important to me… And it’s gone*.P 9

#### 3.2.3. Changed Conditions for Exercise and Physical Activity

The participants were aware of the importance of being physically active to manage their underlying medical condition as well as for their mental health. However, the recommendations on social distancing, closed gyms, and limited access to physiotherapy were perceived as barriers. For some, this meant a reduction in their physical activity levels. Those who used to go to physical therapy found it difficult to exercise on their own and felt that it affected their health negatively.

…*so from the first week in November, I think it was about then that we had more restrictions regarding going to gyms…. So I haven´t gone to the gym since then and I haven´t replaced it by doing something else. I´m moving at lot less now, and that´s actually something that, based on my cardiovascular condition, that I´m actually, eh… worried about*.P 11

*The physiotherapy was also closed. There´s a specialized physiotherapy department for adults suffering from congenital heart diseases. But they had been closed down since March, so that wasn´t an option anymore to go there. //… So that’s a very big difference. So, I notice that I’m weaker in my muscles and stiffer than I was a year ago, I haven’t been able to train that way*.P 5

Others tried to find alternative ways to exercise regularly by taking advantage of outdoor possibilities. Some spent more time outdoors, even when the weather was a bit bad. Those who went to gyms that were still open tried to go very early in the morning or late at night to avoid crowding. 

*I avoid going to classes with lots of people //…I go during such hours, … when I go late, then I go very late, like half nine or nine o´clock at night. And that sort of makes me exhausted in the morning the next day*.P 1

### 3.3. Curbing Emotional Reactions

#### 3.3.1. Dealing with a Shrinking Life-Space

The participants felt that the restrictions affected them both physically and mentally. Daily life had become monotonous and predictable, and there was no longer room for spontaneity. Many activities were shut down, such as going to the hairdresser, the cinema, or the theater, and traveling was difficult, which was felt as a big loss. Some felt that they no longer had a purpose or goal in life and that their life space was shrinking. The restrictions meant that life had become dull, and it felt like living in a void. The participants felt bereft of the good things in life. It was hard to find the silver lining and feel happy. 

*I´m retired and consequently I don´t have a job… I feel like a housewife with my hands tied behind my back, as I can´t shop, I can´t meet people and children and grandchildren, and so I live a very boring life*.P 8

…*we were a whole bunch of people that went to the movies together and then we went to a restaurant, and so on, you can´t do that anymore… we liked to go to the opera, but that has also been put to an end. You can´t go to the library, you just read your old books over and over again*…P 3

Despite these experiences, participants also described how they tried to make the best of their situation. This could entail making an extra fancy dinner at home with one’s spouse instead of going out to restaurants, meeting a few friends for coffee, or enjoying nature.

*But this is compensation, it’s good that you can do it. Last Sunday the weather was nice, so I invited some friends for coffee on the covered terrace of my cabin. We sat there in the sun, at a proper distance, and drank coffee and, uh, ate something. Sponge cake or something like that. So, I’m like that, I’m constantly trying to find “What can I do to keep my spirits up*?”P 6

#### 3.3.2. Applying an Unhealthy Behavior

Different unhealthy behaviors, such as long hours of watching TV series and movies, were described. The behavior led to a sedentary lifestyle, and the intake of fast food to provide comfort and well-being was only effective in the short term. Some also became more careless in taking care of their personal hygiene.

…*I mean, it´s so much easier when you spend so much time at home that you… stuff yourself with all sorts of things. I´ve noticed a big difference there, I haven´t much of a sweet tooth, but I eat more candy now than I´ve done before with chocolate. I feel that there´s often something that slides down my throat when I´ve done my shopping, and I think it´s to comfort myself*.P 1

…*my behavior has changed. Normally I take a shower every day…. But during the pandemic, the personal hygiene has become worse… I rarely shower, like maybe twice a week, and I´m much more careless with my dental hygiene as well*…P 11

#### 3.3.3. Managing Worry and Frustration

The participants used several strategies to manage worry and frustration following the restrictions and their fear of catching COVID-19. Some compared their own situation with people who were much worse off, for example, those who were living in countries with harsh lockdowns and quarantines. 

… *you should be able to manage a situation like this without being depressed. There are others that are in a much worse situation. Think about those who are hit by the pandemic and live in refugee camps, I shouldn´t be complaining*.P 8

Strategies also included seeking information and support, accepting the situation, and not worrying about what they could not control. Others described how they were inventing excuses to stay at home and avoid getting a grip on their situation. 

…*well, I can´t go about and worry before something happens, because then I will have a horrible life. So, what will be will be*.P 1

*It´s been a lot of excuses… blaming something else…eh “Ok, now I can´t make it to the gym, it… it´s better that you respect the recommendations, and…eh… you have a lot to do and it´s bad weather”… eh … lots of excuses to not go out and… and simply take a walk*.P 11

## 4. Discussion

The main findings of this qualitative study were that a majority of the participants were aware of being in an exposed situation. They became worried when faced with the consequences of COVID-19 as their medical condition made them vulnerable. Due to the restrictions, they were no longer able to take part in various physical and social activities, which was hard for them to tackle. Although emotional and psychological distress were present, many participants managed to find strategies that reduced their worries, and they could also perceive positive aspects in everyday life. However, some felt incapacitated and at a loss, and resorted to a more sedentary lifestyle and unhealthy diets. 

### 4.1. Awareness of an Exposed Situation

Some, but not all, identified themselves as belonging to a risk group and tried to comply with the restrictions. However, the daily news being reported was not entirely positive as it contained information regarding the number of people who had died that day. The risk of developing severe pneumonia and hypoxemia by COVID-19, which can lead to severe cardiovascular dysfunction [21,22,24], caused a lot of worry among our participants as there was no telling how hard the virus would strike. Belonging to a risk group has been reported to be associated with higher psychosocial distress [7,25], anxiety, depressive symptoms, and lower life satisfaction, and some of these aspects were also reported among our participants.

Moreover, the stricter recommendations that were given to people > 70 years old and to people with chronic conditions were important in order to decrease the risk of getting contaminated [14]. These restrictions involved avoiding places where people gathered, asking for help in buying food and fetching prescriptions, traveling by public transport, keeping physical distancing, socializing outdoors, frequent handwashing, etc. [6,14]. Although these restrictions were considered important to adhere to among our participants, it was the limited social activities that caused most psychosocial distress, in line with previous studies [7,11,13]. Even short-term periods of social distancing are associated with increased emotional and psychological distress [19].

### 4.2. Changes in Daily Life Habits

Although the restrictions were more lenient in Sweden compared to other countries [6,13,16], they brought considerable changes in daily routines. These changes were perceived differently among those who were still working compared to people who were retired or on sick leave. 

Most workplaces quickly adapted to the new regulations, and those who could work from home were supported by various digital solutions. Meetings were scheduled on various digital platforms, but could not compensate for face-to-face meetings, as spontaneous chats and laughter over a cup of coffee, chats in the corridor, and impulsive lunch meetings with colleagues were sadly missed, according to the participants; this has also been reported previously [26].

The most difficult part of the restriction was maintaining social interaction with family and friends. Not being able to have physical contact and, for example, hug your children and grandchildren aroused both sadness and frustration. A reduction in social contact is associated with higher anxiety, depression, loneliness, and psychosocial distress [7,27]. The most feasible solution among our participants to maintain social interaction was to meet outdoors. This was something they could carry out spontaneously, although it was weather dependent if they wanted to meet friends or family living close by. The most frequent activity was walking together. Arranging a celebration or a lunch required a lot of planning and preparations, and therefore having a barbecue was exceedingly popular, in line with a previous study [28].

The weather could also affect the motivation to exercise outdoors. However, in order to stay healthy, the participants tried to exercise regularly or went to the gym, which was allowed if a distance was kept. Those who were in need of specialized physiotherapy had to manage on their own as these services had closed down, in line with other countries [29,30]. Some managed to continue with physical activity and even increased their outdoor activities, such as taking walks or cycling. Others, especially those in need of specialized care, found it hard to motivate themselves to perform physical activity outdoors. Having access to digital support from specialized health professionals might have been beneficial for our participants, although a few studies indicate that home-based programs by telephone or video are not as effective and motivating as face-to-face contact [29,30]. Moreover, the attitude towards the situation during the pandemic differed among those who were living alone and those who were in a relationship. Living alone aroused feelings of loneliness and worry, which has been reported previously [27,31]. People living in a relationship expressed that they felt fortunate to have a partner to share everyday life with and that it helped them to adapt better, in line with previous studies [17,31].

### 4.3. Curbing Emotional Reactions

An interesting finding was the strategies our participants used to curb the emotional reactions aroused by the life-changing circumstances following the restrictions. At the beginning of COVID-19, the initial reactions were resentment of being regarded as vulnerable, which may have several reasons. Being labeled as weak and vulnerable may be interpreted as being less worthy than others. Previous studies have shown that people with other chronic diseases, for example, late effects of polio and Parkinson´s disease, were striving to do as well as their healthy peers and wanted to be regarded as no different [32,33,34]. This type of minimizing and avoidance was also present among our participants, and similar findings have been described by Kristofferzon et al. [35]. In that study, coping strategies were associated with a sense of coherence and improved mental quality of life. In our study, the emotion-focused coping strategies of minimizing, social comparisons (comparing one´s situation with those worse off), and “we´re all in the same boat” may be comforting strategies [36]. However, our participants also converted emotion-focused coping strategies to problem-focused strategies, e.g., in the spirit of making the best out of the situation, such as arranging a fine dinner, which aroused a positive feeling of well-being. Emotion-focused coping and problem-focused coping may alternate depending on how efficient the strategy is perceived [35]. This was also the case in our study. Some participants slowly resigned to the emotional strain from feeling lonely and that the life space was shrinking, resulting in a more sedentary lifestyle with emotion-focused behaviors, such as spending hours watching TV and eating unhealthy food. Previous studies have found the same patterns of eating and drinking [37,38], which may lead to increased risks of worsening the underlying medical condition. They felt that they had lost the ability to take the initiative and get a grip on their situation. Those who previously had been exercising regularly realized that they were making excuses to avoid going to the gym or taking a walk outdoors, although they were aware of the importance of regular exercise to manage their cardiovascular condition, which is in line with previous studies [39,40].

### 4.4. Strengths and Limitations

The participants were strategically selected and covered a broad range of variations in age, clinical symptoms, and years of various cardiovascular conditions. During the interviews, a semi-structured interview guide was used to ensure that all areas were covered. The 15 participants (nine women) gave rich and varied data from their lived experiences during the pandemic, which gave us a better understanding of their needs and experiences. In the last interview, no new information emerged. However, although we had relevant narratives from all participants, the results are not representative of all people with chronic cardiovascular diseases. Reflexivity was considered during the process of analyzing the data, and we had continuous discussions to keep us aware that our decisions might be influenced by our previous experiences [41]. To add transparency and trustworthiness to our findings and interpretations of the data, we have described the process of analyzing the data and added codes to each quotation to show the representation of our participants.

## 5. Conclusions

From these narratives, we have learned that people with various chronic cardiovascular conditions had to manage not only the threat of catching COVID-19 but also to manage how this stress was affecting their health and well-being. They were cut off from their specialized units that provided support and care, which was worrisome. For some, the situation was no longer manageable and meaningful and was difficult to grasp, which may explain why their coping strategies were no longer efficient. Therefore, they turned to other sources of comfort, such as unhealthy diets and sedentary options. In future crises, and in order to forestall this type of ordeal, support groups with specialized knowledge regarding chronic diseases must be available to help find problem-focused solutions that support self-confidence, self-efficacy, purposeful coping strategies, and a sense of coherence.

## Figures and Tables

**Figure 1 ijerph-20-05588-f001:**
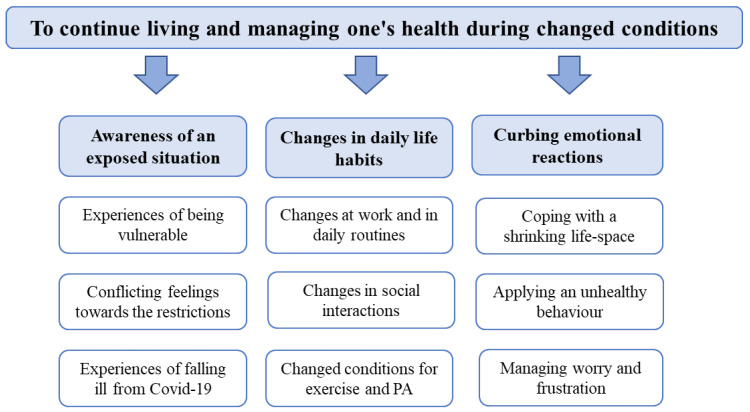
A summary of the theme, categories, and elaborated subcategories.

## Data Availability

Data are available only upon request to the authors, according to the ethical approval from the Swedish Ethical Authority.

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
