# Peer review of "Experiences of People with Cardiovascular Disease during COVID-19 in Sweden: A Qualitative Study"

_ijerph, 2023, doi:10.3390/ijerph20085588_

Round 1
Reviewer 1 Report
The manuscript analyses an interesting topic of patient population group with cardiovascular conditions and their experience regarding their life situation, physical and mental health during the second wave of the pandemic in Sweden. In general, the manuscript is conceptually correctly structured. Introduction gives an insight into the mostly negative effects of significant changes in lifestyle and behavior during Covid-19 pandemic on physical, mental and social component of health at population level. Authors reasonably pointed out that there is lack of relevant information regarding the impact of these changes on vulnerable population groups including patients with chronic cardiovascular conditions. Materials and Methods section is according to my opinion the best part of the manuscript because it depicts quite innovative approach in qualitative analysis that enables the analysis of the results in a very objective way, minimizing the possibility of bias during processing. The results are presented in an appropriate manner and correctly discussed, and logical conclusions were drawn from them. The main objection according to my opinion could be that the results mostly confirmed already known facts, which reduces the originality of the work itself. Literature references are appropriately selected, updated and correctly cited.
Advantages:
- In deed, as authors mentioned there is limited amount of literature data regarding the vulnerable population groups, including population with cardiovascular diseases during the Covid-19 pandemics. Manuscript describes and analyses their experiences, methods of coping with changed living situation as well as the perceived consequences of it on their physical, mental, and social component of health. This feature, I believe, is one of the strengths of the research.
- Authors emphasized in the Conclusions the necessity that, in case of any other similar future crisis, of establishing and enabling support groups with specialised knowledge regarding any chronic disease that should be available to offer specialised support to patients with some chronic disease. That is what I believe is another important message of this manuscript.
Disadvantages:
On the other hand, the number of participants is too small, although the study design and the research was conducted as qualitative research. That, according to my opinion, is one of disadvantages of the manuscript since the participants belong to various diagnostic subgroups of cardiovascular diseases. However, such an approach can be acceptable because even during the second wave of Covid-19 pandemics the vulnerable groups were also defined very roughly with wide range of cardiovascular diagnoses that were included in just one group designated as most vulnerable one.
In conclusion:
Since the study was already performed and it gave certain results that in this stage couldn't be changed, I recommended the acceptance in present form since, according to my opinion, the results are interesting enough and should be shared with wider auditorium.
Reviewer 2 Report
The paper is original and it approaches an important aspect of the life style and clinical problems during the COVID 19 pandemia . The principal limit of the study is the restricted group of the population investigated .
Some peculiar points have been investigated and the session dedicated to the physical activity results particularly important . In this case the authors should clarify some data regarding how the subjects manage the eventual reduction of the intensity of the physical activity and the sleep modifications, if present . In addition if it is possible to define the spontaneous physical activity level regarding the normal daily activity at < 2MET level as walking leisure activities , vs the higher levels by gender .
Did you verify if the life style was influenced by the marital status? .
Some others aspects could be implemented regarding the job carrier expectance and the eventual perception to loose of work opportunities
